

# Serpin functions in host-pathogen interactions

Jialing Bao[1], Guoqing Pan[1], Mortimer Poncz[2,3], Junhong Wei[1], Maoshuang Ran[1] and Zeyang Zhou[1,4]

[1] State Key Laboratory of Silkworm Genome Biology, Southwest University, Chongqing, China
[2] Department of Pediatrics, The Perelman School of Medicine, University of Pennsylvania, Philadelphia, PA, United States of America
[3] Division of Pediatrics, The Children's Hospital of Philadelphia, Philadelphia, PA, United States of America
[4] College of Life Sciences, Chongqing Normal University, Chongqing, China

## ABSTRACT

Serpins are a broadly distributed superfamily of protease inhibitors that are present in all kingdoms of life. The acronym, serpin, is derived from their function as potent serine proteases inhibitors. Early studies of serpins focused on their functions in haemostasis since modulating serine proteases activities are essential for coagulation. Additional research has revealed that serpins function in infection and inflammation, by modulating serine and cysteine proteases activities. The aim of this review is to summarize the accumulating findings and current understanding of the functions of serpins in host-pathogen interactions, serving as host defense proteins as well as pathogenic factors. We also discuss the potential crosstalk between host and pathogen serpins. We anticipate that future research will elucidate the therapeutic value of this novel target.

## INTRODUCTION

Serpins are a superfamily of proteins. The family name is derived from the functional attributes of the members as they are serine protease inhibitors (serpin). They are the most broadly distributed protease inhibitors, and are present in all kingdoms of life including plants, animals, bacteria, archaea and viruses (*Silverman et al., 2001*). While the majority of serpins function as serine protease inhibitors, some serpins function as "cross-class" inhibitors of other kinds of proteases. For example, viral serpin CrmA inhibits a cysteine protease, interleukin-1 beta converting enzyme (*Irving et al., 2002*; *Ray et al., 1992*). In addition, there are a few serpins that exhibit no inhibitory functions but participate in biological processes in other ways. For examples, serpin HSP47 serves as a chaperone, and ovalbumin (another serpin), functions as a storage protein (*Law et al., 2006*).

As potent protease inhibitors, serpins modulate a wide variety of proteolytic cascades thus controlling many physiological and pathological reactions. For instance, human serpins are found to regulate the proteolytic cascade that is central to blood clotting. Antithrombin, a serpin superfamily member, can inhibit multiple key enzymes in blood coagulation such as thrombin, activated factor X (FXa), FIXa and FXIa (*Aguila et al., 2017*;

Corresponding author
Zeyang Zhou, zyzhou@swu.edu.cn

*Hepner & Karlaftis, 2013*). In addition to blood coagulation, serpins also participate in a wide variety of other biological processes. These processes include thrombosis (*Van de Water et al., 2004*), immune-regulation (*Pemberton et al., 1988*), tumour-suppression (*Dzinic et al., 2017*), chromatin condensation (*Grigoryev & Woodcock, 1998*) and apoptosis regulation (*Ray et al., 1992*). Furthermore, studies reveal serpins have clinical relevance. For example, patients with papillary thyroid cancer have high-concentrations of SERPINE2 and SLPI (secretory leukocyte protease inhibitor) (*Stein & Chothia, 1991*). The serum concentrations of both anticoagulant proteins are considered markers for the development of this disease. Kallistatin, another serpin family member, has been shown to regulate cardiovascular function and blood vessel development. Its levels are elevated in patients with type 1 and type 2 diabetes with chronic diabetic complications (*Gateva et al., 2017*).

Recently, the study of serpin functions in infection and inflammation has been of particular interest, especially as more serpins from pathogens are identified and characterized. One example is crmA, a cowpox viral serpin, one of the smallest members of the serpin superfamily. CrmA is thought to be important in allowing viruses to avoid host inflammatory and apoptotic responses (*Ekert, Silke & Vaux, 1999*; *Renatus et al., 2000*). Usually, viral genomes are compact to fit their unique life style. The presence of viral serpins indicates their essential functions for the survival of the pathogen and the infection of hosts. Since serpins are present in both pathogen and host organisms, we will discuss the functions of serpins on each side of these processes, and their potential interactions, in a variety of organisms during infection and inflammation.

## SURVEY METHODOLOGY

In this study, we reviewed articles related to the functions of serpins in different organisms that either serve as hosts or pathogens. All references in this review were retrieved using search engines such as PubMed and Google Scholar. Keywords such as serpin, serine protease, host-pathogen interaction, infection and inflammation were used to search for the references. Figures related to protein structures were searched and modified from the Protein Data Bank.

### Mechanism of serpin inhibition

The mechanism of serpin-inhibiting proteases relies on a reactive center loop (RCL) that interacts with the target enzyme. Structural studies reveal that serpins are characterized by a common core domain consisting of three β-sheets and 8–9 α-helices and a reactive center loop, as shown in Fig. 1A (*Moreira et al., 2014*; *Patschull et al., 2011*; *Pearce et al., 2008*). Although this core region is present in all serpins, sequence homology among members in the family could be as low as 25%. A phylogenetic study of the superfamily divided serpins into 16 'clades', designated clade A-P (*Gettins, 2002*), and includes serpins from vertebrates, invertebrates, plants and viruses. A phylogenetic tree of representative serpins from each clade is shown in Fig. 1B.

Structural studies demonstrated that the RCL region contains a scissile bond between residues P1 and P1′, that interacts with and can be cleaved by the target protease (*Li et al., 1999*). Upon cleavage, the reactive center loop of serpin inserts into the β-sheet A. This

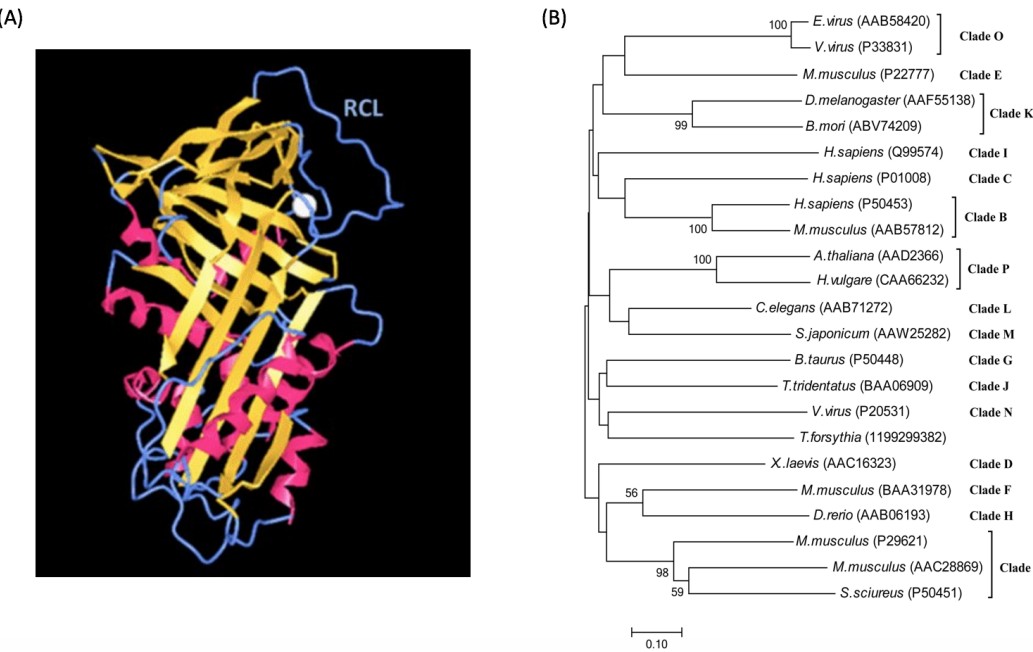

**Figure 1 Serpin structure and phylogenetic tree.** (A) Structure of the serpin alpha-1 antitrypsin. Human alpha-1 antitrypsin is representative of serpin structures. It contains α-helices (red), β-sheets (golden) and a reactive center loop (RCL, the upright blue region) (PDB: 3NE4). (B) Phylogenetic tree of serpin superfamily. The neighbour-joining tree is based on serpin protein sequences and different clades are represented by a single identifier (e.g., Antithrombin III, P01008), where possible. The phylogenetic analysis was performed using MEGA version 7.0. Analysis was done on 1,000 bootstrapped datasets and values of >50% are shown.

conformational change makes serpins thermodynamically more stable (*Dementiev, Dobo & Gettins, 2006*; *Gong et al., 2015*). In addition, a fluorescence study demonstrated that the protease in the complex was also conformationally distorted (*Elliott et al., 1996*). As a result, the target protease is trapped in a covalent and irreversible complex with the serpin, and thus is inhibited irreversibly (*Irving et al., 2000*; *Stein & Chothia, 1991*). This process is illustrated in Fig. 2A, and a structure model showing covalent serpin-protease complex is shown in Fig. 2B.

It is worth noting that co-factors are sometimes needed or can enhance serpins' inhibitory functions. For example, the glycosaminoglycan heparin, a known anti-coagulant, enhances inhibition of cathepsin L by serpin B3 and B4 (*Higgins et al., 2010*). It is also interesting to know that serpins can be secreted or intracellular, thus may also impact their targeted proteases and ways of functions. For instances, the secreted serpins such as SERPINA1 and SEPRINA3 can inhibit inflammatory response molecules; while the intracellular serpins such as SERPINB9 acts on cytosolic proteases thus participate in cellular events (*Law et al., 2006*; *Lomas, 2005*; *Sun et al., 1996*). This does not mean that different forms of serpins have distinct functions, in fact many intracellular serpins participate in inflammatory responses, or vice versa; it is just something we shall keep in mind when discuss the underlying mechanisms of serpin functions as defense factors and pathogenic agents.

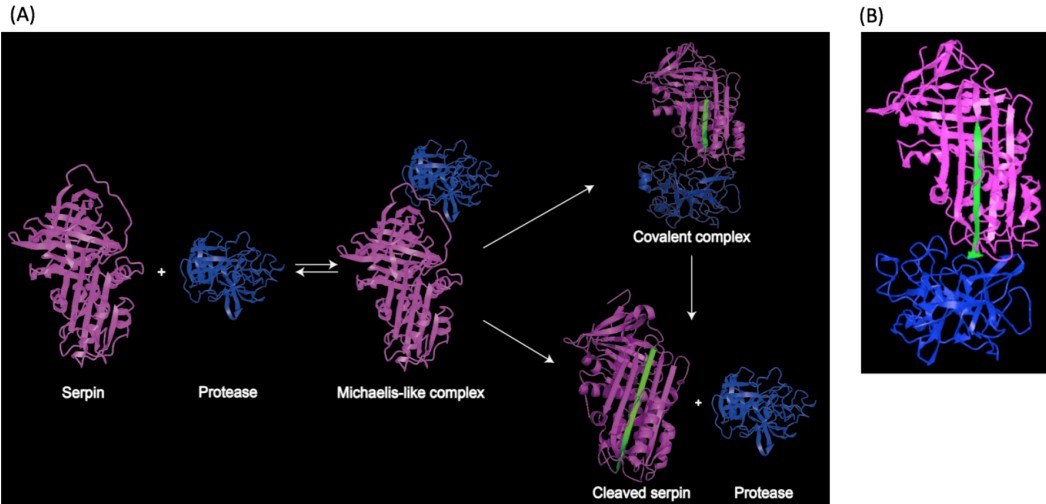

**Figure 2 Representation of serpin-protease interaction.** (A) Proposed process of serpin-protease interaction. A serpin (magenta) interacts with targeted protease (blue), and the Michaelis-like complex of serpin and protease is formed. The complex either undergo peptide bond hydrolysis resulting in a kinetically trapped loop-inserted covalent complex (inhibitory pathway), or a cleaved serpin and free protease (non-inhibitory/substrate pathway). The cleaved and inserted RCL is highlighted in green. Serpin-protease complex is stable. Possibility of transition from covalent complex to cleaved form exists yet slim, since complex *in vivo* would be cleared long before complex decay could occur. (B) Structure of stable serpin- protease complex (PDB: 2D26). The complex is formed by serpin α1PI (magenta) and protease elastase (blue). The inserted RCL is highlighted in green.

## Serpins serve as host defense factors

The defense strategies of serpins derived from the host are variable, including direct inhibition of pathogen proteases, inhibition of pathogen binding, and enhancement of host immune cell functions. Here we discuss mechanisms of how serpins function as host defense factors in a few representative organisms, such as humans, insects and plants.

### Serpins in vertebrates

In humans and other vertebrates, neutrophil extracellular traps (NETs) are web-like DNA structures extruded into the extracellular environment by activated neutrophils. NETs are thought to represent a unique defense strategy against microbial infection. A serpin superfamily member expressed by macrophages and neutrophils is SerpinB1. It is capable of restricting NET production. Studies indicate that SerpinB1 inhibits neutrophil elastase, cathepsin G, and proteinase 3 (*Farley et al., 2012*). More recently, serpins expressed at the mucosal surface have been linked to inhibition of HIV binding, replication and reduction of inflammation of susceptible cells. These serpins, together with other protease inhibitors, are found to be expressed at the epithelial layer of the female genital tract, and thus are considered as essential in the frontline defense against infection. In addition, their potential applications in disease treatment have also been explored (*Aboud et al., 2014*).

Serpins are also found to regulate defense reactions in other mammalian species, such as mouse. A serpin superfamily member α1-antitrypsin promotes lung defense against *Pseudomonas aeruginosa* (PA) infection in mice. A study by *Jiang et al. (2013)* demonstrated

that the underlying mechanism by which α1-antitrypsin reduces lung bacterial infection is through inhibiting neutrophil elastase-mediated host defense protein degradation. Potential therapeutic application of α1-antitrypsin to both humans and mice during PA infection has been proposed.

### Serpins in invertebrates

Serpins have been described in invertebrates, particularly insects. Insects utilize innate immunity as the major defense system against pathogen invasion. The immune responses include hemocyte clotting, melanization and antimicrobial peptide expression (*Meekins, Kanost & Michel, 2017*). To set off these innate responses, cascades of serine proteases activation must be triggered, and these proteolytic cascades are tightly regulated by serpins. Many serpin genes have been identified in species of insects including 34 in *Bombyx mori*, 32 in *Manduca sexta*, 31 in the beetle *Tribolium castaneum* and 29 in *Drosophila melanogaster*. Insect serpins are found expressed in various organs including fat bodies, midgut and hemocytes (*Meekins, Kanost & Michel, 2017*). The majority of these insect serpins are believed to be related to innate immunity. For example, when *Bombyx mori* was challenged by pathogens such as *Micrococcus luteus* and *Serratia marcescens*, the expression of serpin6 (BmSerpin6) was increased significantly (*Li et al., 2017*). It was found that BmSerpin6 directly inhibited the expression of the antimicrobial proteins drosomycin and gloverin2, and the prophenoloxidase (PPO) activity in the melanization cascade.

In studies of *Manduca sexta*, researchers demonstrated that serpin-1 could form a complex with the serine protease hemolymph protease 8 (HP8), to inhibit the activation of the Toll pathway (*An, Ragan & Kanost, 2011*) during bacterial infection. Serpin-7 was found to inhibit prophenoloxidase-activating protease-3 (PAP3) in the melanization pathway to down-regulate innate immune responses (*Suwanchaichinda et al., 2013*). In addition, more recent studies showed that *Manduca sexta* serpin-1, 4, 9, 13 and serpin-3, 5, 6 were all able to complex with pro-hemolymph protease 1 (ProHP1), which is a key proteinase in innate immunity of insects (*He et al., 2017*).

In *Drosophila*, necrotic protein is one of the many serpins that have been related to innate immunity. Necrotic protein inhibits the clip domain of serine protease persephone, and *Drosophila* with necrotic protein mutations constitutively express anti-microbial peptide drosomycin in the Toll inflammatory signaling pathway (*Robertson et al., 2003*).

Thus, most insect serpins negatively regulate innate immunity by inhibiting serine proteases that are essential for immune responses. In addition, several studies revealed that insect serpins could also possess direct anti-pathogen activity upon infection (*Levashina et al., 1999*). For example, serpin protein SRPN6 from *Anopheles gambiae* was highly up-regulated in epithelia immediately after bacterial and parasitic exposures. The AgSRPN6 acts directly on parasite clearance by inhibiting melanization and promoting parasite lysis (*Abraham et al., 2005*).

### Serpins in plants

Serpins are present in almost all land plants (*Roberts et al., 2011*). However, the functions of plant serpins remain to be characterized. *In vitro* studies have demonstrated the protease-inhibiting activities of plant serpins, but the lack of target chymotrypsin-like proteases

within plants suggests that plant serpins may target digestive proteases from invaded pathogens or parasites. In addition, there is abundant accumulation of serpins in plant seeds endosperm, phloem of coleoptiles and leaves (*Roberts et al., 2003*). These localizations of plant serpins further imply the defensive roles of serpins in plants against exogenous proteases and pathogens. Although plant serpins may have distinct characteristics from their insect and animal counterparts, they have been shown to have a role in the pathways regulating the host immune responses. For example, *Yoo et al. (2000)* demonstrated a serpin protein, *Cucurbita maxima* phloem serpin-1 (CmPS), had effective elastase-inhibiting activity. They showed that increased expression of CmPS-1 within the phloem sap was associated with reduced ability of sap-sucking insect aphids to survive and reproduce.

Plants serpins are also found to participate in plant immunity as negative regulators of stress-induced cell death, or so-called hypersensitive response (HR). For instance, *Arabidopsis thaliana* serpins AtSRP4 and AtSRP5 negatively regulate stress-induced cell death induced by bacteria (*Bhattacharjee et al., 2017*). This kind of cell death or response usually occurs at sites where pathogens attempt to invade. Thus the activities of serpins may have a protective role for plants when facing pathogen attack.

In addition to the above mechanisms of serpins acting on proteases to function as host defense factors, there are also other ways that serpins participate in host defense. For example, serpins can induce the expression of host antimicrobial peptides and cytokines (*Kausar et al., 2017*; *Zhao et al., 2014*). Serpins can also directly bind to bacterial pathogens and cause membrane disruptions (*Malmstrom et al., 2009*). Interestingly, even non-inhibitory serpin can exert antibacterial activity. For example, non-inhibitory serpin ovalbumin-related protein X possesses antibacterial activity through heparin-binding ability (*Rehault-Godbert et al., 2013*). All these findings broaden our understanding of the mechanisms of serpin functions as host defense factors.

## Pathogen-derived serpins in infection and inflammation

As mentioned above, serpins are present in almost all kingdoms of life including microbes and other pathogenic organisms. Pathogen-derived serpins may facilitate infection or survival of pathogens, but the mechanisms remain to be fully elucidated. The data indicate that pathogen-derived serpins are capable of inhibiting host inflammatory proteins or cells, and abrogating host cell apoptosis. Studies are on-going to find additional strategies used by pathogen-derived serpins to facilitate infection. In addition, the potential of pathogen-derived serpins as novel candidates of clinical therapies or vaccination has drawn great interest from scientists and physicians.

### Serpins from viruses

Viral genomes are usually kept at minimum scales to fit their unique life style. Thus the presence of viral serpins must be essential for the survival and/or function of the virus. In fact, researchers have identified several serpins that are required for virulence and infectivity (*Nathaniel et al., 2004*). In the myxoma viruses, three serpins, SERP1, SERP2 and SERP3 have been identified (*MacNeill, Turner & Moyer, 2006*). Similarly, there are three serpins in orthopox viruses, designated SPI-1, SPI-2 and SPI-3 (*Macen et al., 1996*).

In addition, the P1 positions of SPI-2-like and SERP2 serpins contain an aspartyl residue, which indicates their potential targets are mammalian caspases and the serine protease granzyme B (*Turner et al., 1999*). By inhibiting these host proteases, the virus may be able to restrain apoptosis in host cells thus down-regulatinghost immune responses. Other viruses have also been shown to contain one or more serpins, including swinepox, lumpy skin disease virus, fowlpox, and members of the rhadinovirus genus. Recently, a baculovirus serpin Hesp018 has been identified in the *Hemileuca* species nuclear polyhedrosis virus (NPV). This serpin protein has been suggested to abrogate host cell apoptosis, resulting in accelerated production of virus in Sf9 insect cells (*Ardisson-Araujo et al., 2015*).

The ability of viral serpins to abrogate host immune systems has been proposed as a strategy to treat certain diseases. One such example is the proposed application of the viral serpin Serp-1 to treat acute unstable coronary syndromes (*Lucas et al., 2009*). Several other viral serpins are being studied for their potential applications as novel anti-inflammatory therapeutics as well (*Mangan et al., 2017*; *Zheng et al., 2012*).

### Serpins from bacteria

*Tannerella forsythia* is an anaerobic, gram-negative bacteria species that usually resides in the human mouth and contributes to chronic periodontitis (*Pereira et al., 2017*). To inhibit host endopeptidases, *T. forsythia* secretes a serpin-type protease inhibitor called miropin, which irreversibly inhibits serine and cysteine endopeptidases of the host (*Goulas et al., 2017*). Phylogenetic analysis of this serpin protein shows that it does not follow a vertical descent model, indicating micropin may arise from the host by horizontal gene transfer. In fact, the studies of serpins from human commensal bacteria and their therapeutic applications have become exciting research areas. For instance, *Mkaouar et al. (2016)* characterized two novel serpins from the human gastrointestinal tract commensal bacteria. These two serpins are called siropin-1 and siropin-2. These two serpins are found to preferentially inhibit two human serine proteases, neutrophil elastase and proteinase 3, that are associated with human inflammatory bowel disease (*Mkaouar et al., 2016*). Thus, siropins or other serpins from human commensal bacteria have been suggested as novel therapeutics against human inflammatory diseases.

### Serpins from parasites

Parasites are organisms that live in another organism and may cause major public health problems such as zoonotic diseases. Usually the parasites need to evade host defense system in order to survive. Serpins are found to be important during host-parasite interaction, and parasites utilize their serpins to facilitate infection and survival in the host. For example, cattle tick *Rhipicephalus microplus* encodes at least 24 serpins, of which RmS-3, RmS-6, and RmS-17 were identified in the saliva and later confirmed to inhibit pro-inflammatory and pro-coagulatory proteases of the host (*Tirloni et al., 2014*). In a follow-up study, rRmS-3 was found to inhibit chymotrypsin, cathepsin G, and pancreatic elastase. Among these serpins, rRmS-6 was found to inhibit trypsin, chymotrypsin, factor Xa, factor XIa and plasmin; while rRmS-17 inhibited trypsin, cathepsin G, chymotrypsin, plasmin and factor XIa (*Tirloni et al., 2016*). This study also claimed that polyclonal antibodies to saliva proteins of *Amblyomma americanum, Ixodes scapularis* and *Rhipicephalus sanguineus* were

able to cross-react with these three *R. microplus* saliva serpins. These findings suggest serpins from pathogens could be applied as novel candidates of vaccination (*De la Fuente et al., 2007*).

Researchers have identified several serpins associated with *Trichinella spiralis* including Tsp03044 and TspAd5 (*Knox, 2007*; *Zhang et al., 2016*). Both of them inhibit trypsin, α-chymotrypsinand pepsin of mammals. These data support the inference that serpins from parasites facilitate invasion into host tissues (*Zhang et al., 2016*).

Similarly, *Schistosoma mansoni* has at least eight serpins. Among those, Smpi56 and SmSPI have been characterized. Smpi56 was purified from extracts of adult *S. mansoni*, and is able to inhibit neutrophil elastase, pancreatic elastase and an endogenous cercarial protease (*Ghendler, Arnon & Fishelson, 1994*; *Quezada & McKerrow, 2011*). SmSPI, also known as *S. mansoni* serpin isoform 3, was found to inhibit chymotrypsin, neutrophil elastase and porcine pancreatic elastase (*Pakchotanon et al., 2016*). In addition, SmSPI was found predominantly expressed in the head gland of the parasite, as well as on its spines. These findings suggest that serpins from *S. mansoni* facilitate intradermal and intravenous survival of this pathogen.

## Potential crosstalk between host and pathogen serpins

As we discussed above, under most circumstances serpins target proteases to modulate inflammatory responses during host-pathogen interactions. Yet it is still of great interest to know whether serpins function through serpin-serpin interactions. In fact, there is evidence that serpins can be inactivated through polymerization (*Gettins & Olson, 2016*). Serpin polymerization occurs when the RCL region of one serpin docks into the β-sheet A of another serpin to form an inactive serpin polymer. Actually, this kind of serpin inactivation is not uncommon and many examples are found in human deficiency and diseases. For example, the Z allele of SERPINA1 accumulates in patients' liver through serpin polymerization (*Lomas et al., 1992*). Thus, we would not be surprised to find out in the future that pathogen serpins utilize this mechanism to inactivate host serpins, or vice versa, during infection.

## CONCLUSIONS

Almost all organisms express serpins, and serpins play critical roles in host-pathogen interactions and regulation of inflammatory responses. The evidence indicates that serpins of the host provide protection and those of the pathogens enhance infectivity. A brief summary of representative serpin functions during host-pathogen interactions is shown in Fig. 3.

On the basis of existing studies, we conclude that serpins, either from hosts or pathogens, modulate inflammatory responses by inhibiting target proteases associated with host-pathogen interactions. There is also evidence indicating potential cross interactions between host and pathogen serpins. In addition to all that, there are a number of studies demonstrating that serpin-proteases complexes can lead to downstream signals that result in responses such as inflammation, cytoskeleton rearrangement, proliferation and apoptosis. For example, R1-Anti-chymotrypsin-cathepsin G complexes have been shown

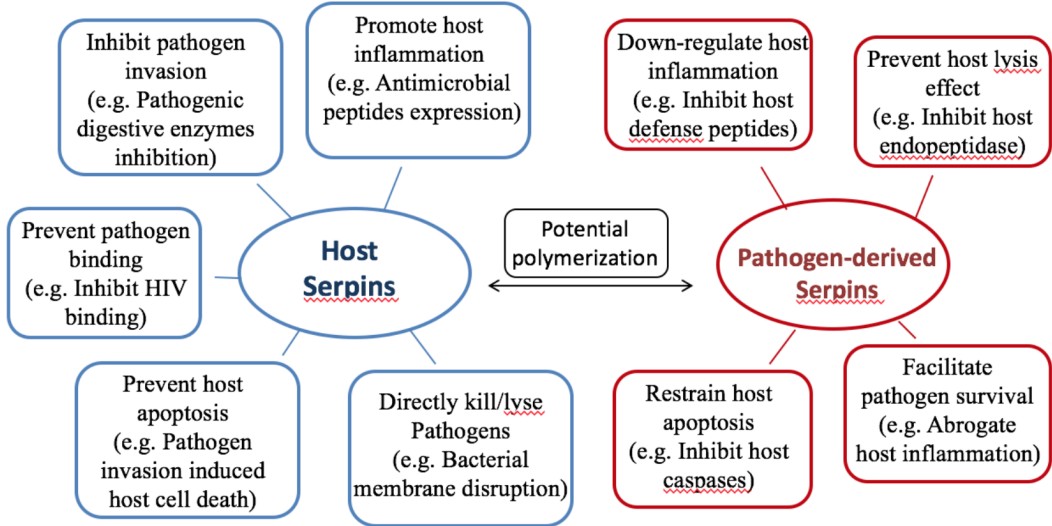

**Figure 3** **Summary of serpin functions in host-pathogen interactions.** Hypothesis of protective mechanisms offered byhost serpins (on the left, blue), and pathogenic mechanisms exerted by pathogen-derived serpins (on the right, red). Host serpins may act directly or indirectly upon pathogen infections. The representative mechanisms include inhibiting pathogenic digestive proteases, promoting host antimicrobial peptide expression and so on. Pathogen-derived serpins also utilize various mechanisms and representative ones are listed.

to stimulate production of interleukin 6 and activation of NADPH oxidase (*Kurdowska & Travis, 1990*; *Schuster et al., 1992*). More recently, serpin α-1 antitrypsin was found to be able to form complexes independently of the inhibitory site with neutrophil-expressedpro-inflammatory leukotriene B4. The resulting complexes modulate downstream signaling events, and augmentation of serpin α-1 antitrypsin was suggested as a potential therapy for inflammatory diseases (*O'Dwyer et al., 2015*). These signaling properties of serpin-protease complexes shed light on the versatility of serpins modulating inflammatory responses. Another interesting yet poorly explored aspect is the activities of the peptide derivatives resulting from proteolytic cleavage of the reactive center loop (RCL) of serpins. A few studies have demonstrated that the peptides derived from serpin RCL have expanded functions such as anti-inflammatory and antimicrobial activities (*Ambadapadi et al., 2016*; *Andersson et al., 2004*). Further explorations are needed for additional activities and potential applications of these peptide derivatives.

There is no doubt that further studies of serpins will identify many more biological targets and underlying molecular mechanisms. The study of serpins will remain an important area for basic research, as well as for clinical applications.

## ACKNOWLEDGEMENTS

We appreciate Dr. Judith S. Bond, Evan Pugh Professor Emeritus, Department of Biochemistry and Molecular Biology Penn State University College of Medicine at Hershey; Adjunct Professor, Department of Biochemistry & Biophysics, University of North Carolina

School of Medicine at Chapel Hill, for her great support for correcting the language in this manuscript.

### Funding
This work was supported by the Fundamental Research Funds for the Central Universities (No. XDJK2017B002) and the National Natural Science Foundation of China (No. 31470250, No. 31472151). The funders had no role in study design, data collection and analysis, decision to publish, or preparation of the manuscript.

### Grant Disclosures
The following grant information was disclosed by the authors:
Fundamental Research Funds for the Central Universities: XDJK2017B002.
National Natural Science Foundation of China: 31470250, 31472151.

### Competing Interests
The authors declare there are no competing interests.

### Author Contributions
- Jialing Bao conceived and designed the experiments, performed the experiments, analyzed the data, contributed reagents/materials/analysis tools, prepared figures and/or tables, authored or reviewed drafts of the paper, approved the final draft.
- Guoqing Pan, Junhong Wei and Zeyang Zhou authored or reviewed drafts of the paper, approved the final draft.
- Mortimer Poncz and Maoshuang Ran prepared figures and/or tables, authored or reviewed drafts of the paper, approved the final draft.

### Data Availability
The research in this review article did not generate any data or code.

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
