# Peer review of "Serpin functions in host-pathogen interactions"

_PeerJ, doi:10.7717/peerj.4557_

## Round 0.1 · original submission · Minor Revisions

Both referees suggest that a more sophisticated or detailed figure (or perhaps an additional figure) would enhance the paper.

Authors should comment briefly on the additional points raised by referee 2 (comments under the headings "Mechanism of serpin action" and "conclusions".

Authors should also try to eliminate the minor typographical and grammatical errors mentioned by both referees.

A phylogenetic tree, mentioned by referee 1 would be desirable but not essential

Reviewer 1 ·

Basic reporting

The review article by Bao et al. is a timely and comprehensive summary on the serine protease inhibitor family of serpins. The authors particularly focus on the functions of these molecules in host-pathogen interactions, which make this review unique and interesting for the field.
However, prior publication some issues should be addressed.

Experimental design

n.a.

Validity of the findings

n.a.

Additional comments

- A phylogenetic tree of different serpins from different species reported in this manuscript should be presented.
- The cartoon in Fig. 2A should be improved. Besides the rather simple scheme a more sophisticated presentation of molecular structures explaining the inhibitory mechanism is needed.
- The text contains many errors and typos, which should be corrected (e.g. line 94: delete ‘of’; line 103: ‘A’ study by …; line 125: ‘bacterial’; line 150: ‘Cucurbita’; line 154 ‘found to’; line 156 ‘thaliana serpins’; line 160: ‘and Inflammation’; line 178: ‘may be able’; etc. etc..

Reviewer 2 ·

Basic reporting

The article by Bao et al. is clear and well written. The subject of the article is rather interesting and it provides an interesting overview of the role of serpins in immunity of both vertebrates and invertebrates. Overall, the article is well-referenced. The article contains only one figure illustrating a common serpin 3D structure from the archetype a1-Antitrypsin. I suggest to replace this structure by a figure showing the interaction of the serpin with its targeted protease and the authors should also mentioned that serpin can either acts as a substrate (resulting in a cleaved inactive serpin) or a covalent inhibitor (refer to Silverman et al., JBC 2001). The author should provide a graphical representation that summarizes serpins’ functions in immunity separating the host on one hand and the infectious agents on the other hand.
Minor comments: I haven’t noticed major grammatical errors but many spaces between two adjacent words are missing (line 80 aresult, line 154, foundto, line 160, serpinsin andinflammation etc.). I do not known whether these errors are in the initial version or only in the uploaded version.

Experimental design

The authors used Pubmed and Google scholar tools and serpin, serine protease, host-pathogen interaction infection and inflammation as keywords. They might have missed some interesting data that should be found using antibacterial or antimicrobial as additional keywords. Indeed, this review is mainly focused on how serpins can regulate host or on the other hand, pathogen proteolytic activities but some serpins have been described to have antibacterial activities independently of their inhibitory site.

Validity of the findings

The finding are rather interesting but as mentioned in 1., the clarity of the message would be improved with a figure summarizing major findings. Some information is still missing (see below).

Additional comments

Mechanism of serpin inhibition:
Please indicate the presence of a noninhibitory pathway that may influence the stoichiometry of inhibition and that depends on the couple serpin/protease considered. Please indicate that some serpins are secreted and some are intracellular which may also have some impact on their targeted proteases and depending on the mechanisms underlying the pathogenicity of infectious agents. Also, indicate that some co-factors are sometimes needed such as glycosaminoglycans including heparin for HCII, PCI, ATIII for example, knowing that heparin is secreted by mastocytes, which are involved in response to infection and inflammation. Please indicate that the mechanism of action of host serpins against pathogens is usually indirect (through the inhibition of microbial proteases which are major virulent factors that trigger dissemination and invasiveness of pathogens) but it can also have a direct effect (interaction and subsequent lysis of pathogens).
Please include a paragraph in this section on other effects (induction of expression of host antimicrobial peptides, cytokines)
Below are some suggested articles:
-Serpin-14 negatively regulates prophenoloxidase activation and expression of antimicrobial peptides in Chinese oak silkworm Antheraea pernyi. Kausar S, Abbas MN, Qian C, Zhu B, Sun Y, Sun Y, Wang L, Wei G, Maqsood I, Liu CL Dev Comp Immunol. 2017 Nov;76:45-55. doi: 10.1016/j.dci.2017.05.017. Epub 2017 May 22.
-Antibacterial activity of serine protease inhibitor 1 from kuruma shrimp Marsupenaeus japonicus. Zhao YR, Xu YH, Jiang HS, Xu S, Zhao XF, Wang JX. Dev Comp Immunol. 2014 Jun;44(2):261-9. doi: 10.1016/j.dci.2014.01.002. Epub 2014 Jan 9.
-Protein C inhibitor--a novel antimicrobial agent. Malmström E, Mörgelin M, Malmsten M, Johansson L, Norrby-Teglund A, Shannon O, Schmidtchen A, Meijers JC, Herwald H. PLoS Pathog. 2009 Dec;5
-Ovalbumin-related protein X is a heparin-binding ov-serpin exhibiting antimicrobial activities.
Réhault-Godbert S1, Labas V, Helloin E, Hervé-Grépinet V, Slugocki C, Berges M, Bourin MC, Brionne A, Poirier JC, Gautron J, Coste F, Nys Y. J Biol Chem. 2013 Jun 14;288(24):17285-95. doi: 10.1074/jbc.M113.469759. Epub 2013 Apr 24.

Conclusion
Please include one or two sentences about functions related to peptides derived from serpin RCL and others that may have some additional bioactivities including some related to the modulation of the response to infections (antibacterial peptides and immunomodulatory activities). Indeed, the interaction of the serpin with its targeted protease leads to a covalent bipartite complex together with the release of a peptide of about 50 amino-acid residues resulting from the cleavage of the RCL and the activities of these peptides have been poorly explored.
Reactive Center Loop (RCL) Peptides Derived from Serpins Display Independent Coagulation and Immune Modulating Activities. Ambadapadi S, Munuswamy-Ramanujam G, Zheng D, Sullivan C, Dai E, Morshed S, McFadden B, Feldman E, Pinard M, McKenna R, Tibbetts S, Lucas A. J Biol Chem. 2016 Feb 5;291(6):2874-87. doi: 10.1074/jbc.M115.704841. Epub 2015 Nov 30.
Antimicrobial activities of heparin-binding peptides. Andersson E, Rydengård V, Sonesson A, Mörgelin M, Björck L, Schmidtchen A. Eur J Biochem. 2004 Mar;271(6):1219-26.

---

## Round 0.2 · accepted · Accept

Thank you for submitting the revised version, with additional figures